# Cerebellar Purkinje cell activity modulates aggressive behavior

Skyler L Jackman[1,2], Christopher H Chen[1], Heather L Offermann[1], Iain R Drew[1], Bailey M Harrison[1], Anna M Bowman[2], Katelyn M Flick[1], Isabella Flaquer[1], Wade G Regehr[1]*

[1]Department of Neurobiology, Harvard Medical School, Boston, United States; [2]Vollum Institute, Oregon Health and Science University, Portland, United States

**Abstract** Although the cerebellum is traditionally associated with balance and motor function, it also plays wider roles in affective and cognitive behaviors. Evidence suggests that the cerebellar vermis may regulate aggressive behavior, though the cerebellar circuits and patterns of activity that influence aggression remain unclear. We used optogenetic methods to bidirectionally modulate the activity of spatially-delineated cerebellar Purkinje cells to evaluate the impact on aggression in mice. Increasing Purkinje cell activity in the vermis significantly reduced the frequency of attacks in a resident-intruder assay. Reduced aggression was not a consequence of impaired motor function, because optogenetic stimulation did not alter motor performance. In complementary experiments, optogenetic inhibition of Purkinje cells in the vermis increased the frequency of attacks. These results suggest Purkinje cell activity in the cerebellar vermis regulates aggression, and further support the importance of the cerebellum in driving affective behaviors that could contribute to neurological disorders.

## Introduction

Profound motor deficits such as ataxia and loss of oculomotor control are the most obvious manifestations of cerebellar damage. This has contributed to the popular view that the cerebellum is involved primarily in motor function, but this is far from a complete view of the behavioral functions of the cerebellum. fMRI studies suggest that some regions of the cerebellar cortex are devoted to motor function, but other regions are involved in working memory, language, emotion, executive function and many other nonmotor functions (*Stoodley and Schmahmann, 2009*; *Van Overwalle et al., 2014*). The cerebellum is also implicated in autism spectrum disorder (*Wang et al., 2014*), anxiety (*Moreno-Rius, 2018*), attention deficit disorder (*Berquin et al., 1998*), schizophrenia (*Andreasen and Pierson, 2008*), and other nonmotor neurological disorders (*Phillips et al., 2015*).

The posterior vermis region of the cerebellar cortex is particularly intriguing with regard to involvement in nonmotor behaviors. Damage to the cerebellar vermis in adults can lead to deficits in executive function, spatial cognition, linguistic processing, affect regulation, irritability, anger, aggression, and pathological crying or laughing (*Schmahmann and Sherman, 1998*; *Levisohn et al., 2000*). There is also extensive evidence suggesting that the vermis influences aggression. In seminal studies mapping the somatotopic organization of the cerebellar cortex, the Italian physiologist Guisseppe Pagano found that injecting curare into the vermis caused the animal to 'become suddenly furious, and throw itself at those present, trying to bite them' or to 'jump into the air, struggling to bite who knows how many phantoms of its agitated psyche' (*Pagano, 1904*). Later lesions studies demonstrated that resection of the vermis had the opposite influence on behavior and produced a calming effect (*Sprague and Chambers, 1959*; *Peters and Monjan, 1971*; *Berman et al., 1974*). Electrical stimulation of the deep cerebellar nuclei has been shown to drive aggressive behaviors such as sham rage (*Zanchetti and Zoccolini, 1954*) and attack (*Reis et al., 1973*). In human clinical

*For correspondence:
wade_regehr@hms.harvard.edu

Competing interests: The authors declare that no competing interests exist.

studies, stimulating the surface of the vermis improved emotional control and reduced aggressive outbursts (*Heath, 1977*), and reduced feelings of anger (*Cooper et al., 1976*).

These studies implicated the vermis in regulating aggression, but they have been difficult to interpret. Purkinje cells (PCs), the sole output cells of the cerebellar cortex, fire continuously at up to 100 Hz, and inhibit neurons in the deep cerebellar nuclei (DCN) that in turn influence other brain regions. The observations that electrical stimulation of the vermis reduced aggression (*Heath, 1977*), while stimulation of the DCN increased aggression (*Reis et al., 1973*), are consistent with PC activity inhibiting cells in the DCN (*Telgkamp and Raman, 2002*; *Alviña et al., 2008*). This leads to the interpretation that elevated PC activity and decreased DCN activity suppress aggression, and conversely elevated DCN activity (and by implication decreased PC activity) suppresses aggression. However, it is difficult to reconcile stimulation studies with the taming effect observed in lesion studies (*Sprague and Chambers, 1959*). Lesions of the vermis will decrease PC inhibition of the DCN, and would thus be expected to promote aggression by elevating firing within the DCN. Instead, they suppress aggression.

Numerous factors make both the electrical stimulation and lesion experiments difficult to interpret, and could contribute to the apparent discrepancy between these two approaches. Electrical stimulation of the cerebellar cortex activates all types of neurons in the vicinity of the electrode, including PCs. Molecular layer interneurons will also be activated, and they inhibit PC firing (*Heiney et al., 2014*). Stimulation also antidromically activates mossy fibers, climbing fibers, and modulatory inputs from other regions (*Llinás and Sasaki, 1989*; *Baker and Edgley, 2006*), which might contribute to the behavioral consequences of stimulation. In addition, electrical stimulation of PCs can synchronize activity, which could paradoxically increase spiking in the DCN (*Person and Raman, 2012a*; *Person and Raman, 2012b*). For these reasons it is unclear how electrical stimulation of the vermal cortex influences cerebellar circuitry. For lesion studies of the cerebellar vermis, it is not clear how the firing of downstream DCN neurons was altered. Although the loss of inhibition would be expected to increase DCN firing, genetic ablation of PCs has been shown to paradoxically decrease firing in the DCN (*Bäurle et al., 1997*), suggesting that compensatory mechanisms regulate the properties of DCN neurons in the absence of PC input. In addition, previous studies did not quantify the magnitude and time course of behavioral effects driven by cerebellar interventions.

To determine the role of the cerebellum in regulating aggression, we used optogenetic techniques to selectively control PC activity in mice. We used the resident-intruder assay, a measure of natural territorial aggression in rodents, to provide a measure of the extent and time course of an aggressive behavior (*Koolhaas et al., 2013*). Manipulating PC firing in the vermis, but not in another cerebellar region, enabled rapid, bidirectional control of aggression. This study provides evidence that in the cerebellar vermis elevated PC firing suppresses aggression, whereas suppressing PC firing promotes aggressive behavior.

## Results

To selectively modulate the activity of PCs in the cerebellar vermis, we used PCP2-cre mice (*Zhang et al., 2004*) to restrict expression of the microbial opsins ChR2 (channelrhodopsin-2, [*Boyden et al., 2005*]) and NpHR3.0 (halorhodopsin, [*Zhang et al., 2007*]) to PCs. In vitro electrophysiological recordings (*Figure 1a*) confirmed that ChR2 stimulation could drive graded increases in PC firing that scaled with light intensity (*Figure 1b*), as previously reported (*Guo et al., 2016*). Similarly, halorhodopsin stimulation decreased PC firing rates in a light intensity-dependent manner (*Figure 1c*). To manipulate PC activity in vivo, optical fibers were chronically implanted in adult (>P42) male PCP2-cre::ChR2 or PCP2-cre::NpHR mice over the surface of the cerebellar vermis. Fibers were positioned at the midline over lobule VII (*Figure 1—figure supplement 1*), a region suggested to play a role in emotional processing (*Stoodley and Schmahmann, 2009*). Subsequent in vivo recordings through an adjacent craniotomy in anesthetized animals confirmed the ability of light from the implanted optical fiber to reliably increase or decrease the firing rate of putative PCs expressing ChR2 or halorhodopsin, respectively (*Figure 1d–f*). As with electrical stimulation, optogenetic stimulation has the potential to induce pauses in PC firing. If pauses occur synchronously among multiple PCs that provide inhibition to a DCN neuron, it could promote firing in the DCN (*Person and Raman, 2012a*). We examined the timing of optically-evoked firing of PCs and found that there were complex temporal responses in different PCs, although there was not a pause in

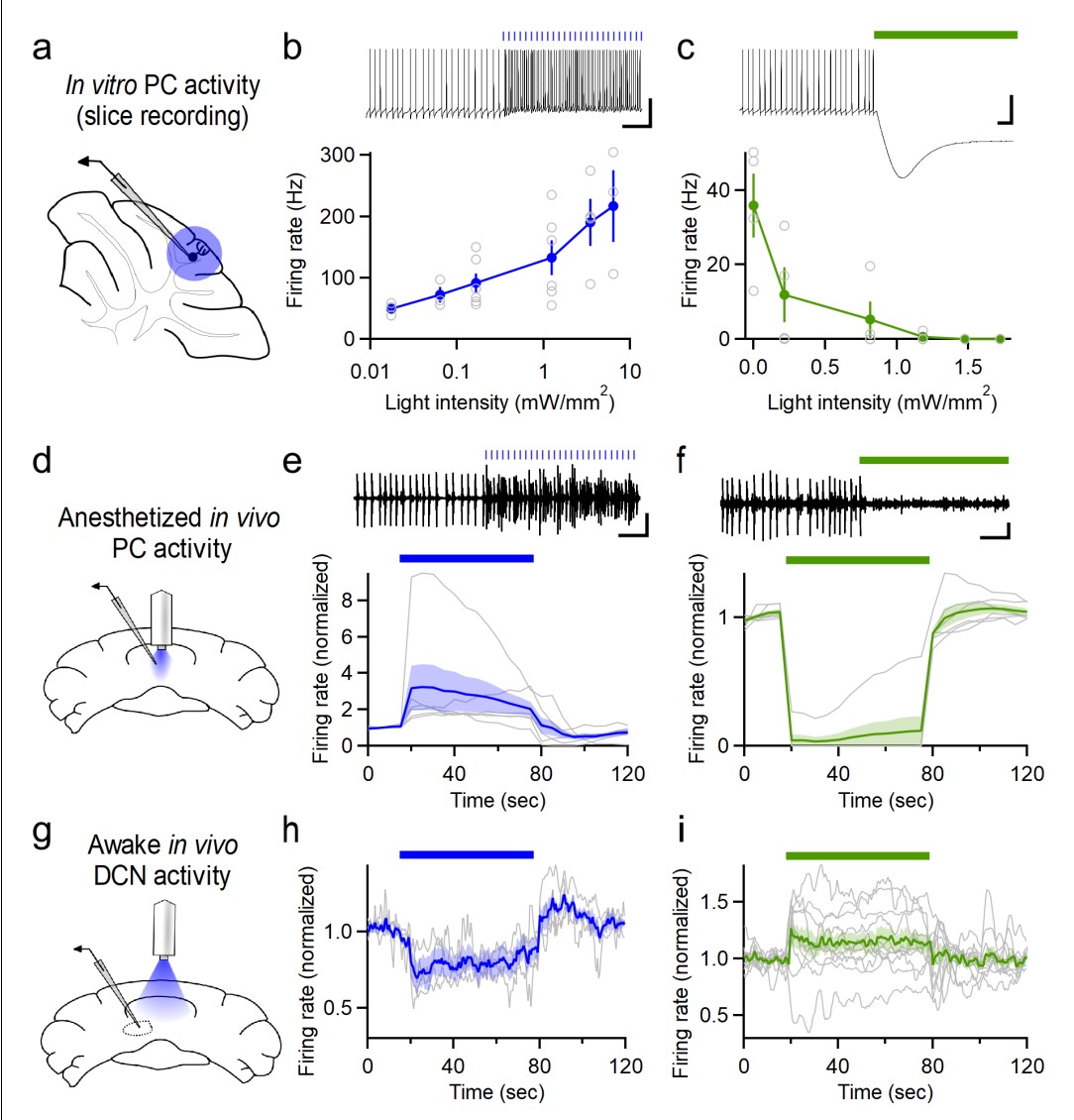

**Figure 1.** Optogenetic control of Purkinje cell activity. (**a**) Recording schematic for in vitro recording and optogenetic stimulation. (**b**) Firing rates elicited by ChR2 stimulation at different intensities (0.5 ms flashes, 50 Hz, n = 6). (**c**) Inhibition of PCs at different light intensities (sustained illumination, n = 4). (**d**) Schematic for recording PC activity during in vivo stimulation through a chronic fiber optic implant. (**e**) Top: Representative single unit recording during ChR2 stimulation and (bottom) average firing rate (n = 6). (**f**) Single unit recordings during halorhodopsin stimulation (n = 6). Scale bars, 100 ms (horizontal), 20 mV (vertical, (**b,c**), 0.2 mV (vertical, (**e,f**). (**g**) Schematic for recording DCN activity during in vivo stimulation in awake animals. (**h**) Averaged normalized firing rate of DCN neurons during ChR2 stimulation of vermal PCs (n = 4). (**i**) Averaged firing of DCN neurons during halorhodopsin-mediated inhibition of vermal PCs (n = 11). Average data in all figures represents mean ± SEM.

The online version of this article includes the following source data and figure supplement(s) for figure 1:

**Source data 1.** Optogenetic firing modulation data.

**Figure supplement 1.** Fluorescent images of ChR2-YFP expression in (a) a whole brain and (b) a sagittal cerebellar section from a PCP2-cre::Ai32 mouse, with lobules V-X labeled.

**Figure supplement 2.** Millisecond-scale analysis of firing rates induced by in vivo stimulation in PCP2-cre::Ai32 animals.

average firing of PCs (*Figure 1—figure supplement 2*). We went on to record responses of neurons in the deep cerebellar nuclei (DCN) to optogenetic manipulations of PC firing in awake animals (*Figure 1g*). Increasing PC firing with ChR2 decreased the firing of DCN neurons (*Figure 1h*, *Figure 1—figure supplement 2*), while decreasing PC firing with halorhodopsin increased the firing of neurons in the DCN (*Figure 1i*). The magnitudes of the changes in DCN neuron firing were relatively

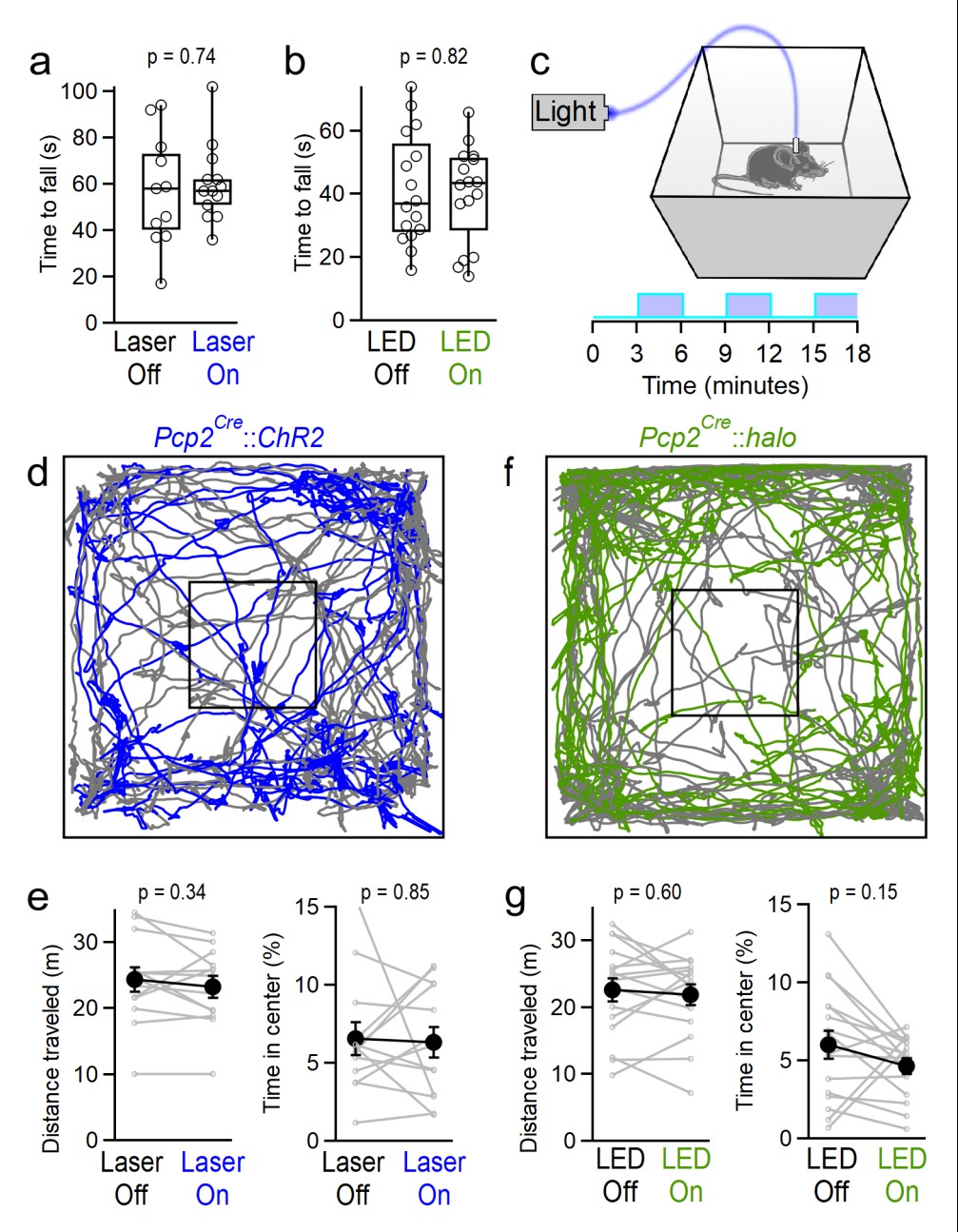

**Figure 2.** Manipulating vermal Purkinje cell activity does not affect coordination, locomotion or anxiety. (**a**) Time to fall for rotarod assays during ChR2-mediated excitation. Mice were tested in two consecutive trials, and randomly assigned to receive stimulation during either the first or second trial. (n = 13) (**b**) Same as in (**a**), but for halorhodopsin-mediated inhibition of vermal PC firing. (n = 16) (**c**) Schematic for open field assay with optogenetic stimulation. (**d**) Representative tracking data throughout alternating periods with stimulation (blue) and without (gray). (**e**) Total distance traveled and time spent in the center of the arena for mice during epochs with and without stimulation of vermal PCs (n = 13). (**f, g**), Same as **d**), **e**) but for halorhodopsin-mediated inhibition of vermal PCs (n = 17).

The online version of this article includes the following source data and figure supplement(s) for figure 2:

**Source data 1.** Open field (ChR2) data.
**Source data 2.** Open field (Halo) data.
**Source data 3.** Rotarod data for *Figure 2A*.
**Figure supplement 1.** Manipulating vermal Purkinje cell firing does not affect locomotion.

modest. This may reflect in part the heightened firing rates of neurons in awake animals. In addition, it was a challenge to record from DCN neurons that were targets of the PC population that was influenced with light and other neurons in the DCN may be more strongly targeted and undergo larger changes in activity than those reported here. For this reason the effects on DCN firing should be considered lower bounds.

After allowing at least 1 week for recovery from fiber implantation surgeries, animals were placed in an open field arena and stimulated with increasing light intensities to determine if manipulating PC activity produced overt motor deficits or behavioral consequences. In ChR2-expressing mice, strong vermal stimulation using the highest light intensities (~110 mW/mm$^2$ at the face of the fiber optic implant) often resulted in clear motor effects, causing mice to become immobile or exhibit seizure-like and dystonic activity. This behavior resembled previous descriptions of seizure-like activity driven by strong electrical stimulation of the vermal cortex (*Clark, 1937*; *Chambers, 1947*). Thus, for all subsequent assays we tailored the intensity of light delivered to each animal to the maximal intensity where animals remained mobile in the open field arena and did not display signs of motor impairment (mean intensity 90 ± 3.9 mW/mm$^2$). In contrast, halorhodopsin-expressing animals exhibited no obvious behavioral effects in response to stimulation at the maximal light intensity deliverable by the fiber-coupled LED light source (61 mW/mm$^2$). This value was used for all subsequent assays.

Because the cerebellum plays well-established roles in motor control and balance, and optogenetic manipulation of PCs in the simplex lobe has been shown to perturb forelimb movements (*Lee et al., 2015*), we first tested whether manipulating vermal PC firing caused more subtle motor impairments than those described above, that might interfere with the expression of other behaviors. To evaluate coordination, animals were tested on accelerating rotarod assays for two consecutive trials during which they received either optical stimulation or no stimulation. Neither excitation with ChR2 (*Figure 2a*) nor inhibition with halorhodopsin (*Figure 2b*) affected the rotarod performance. We next assessed locomotion during open-field assays. Animals received alternating 3 min blocks of optogenetic excitation (*Figure 2c*). Automated animal tracking (*Figure 2d*) revealed that optogenetic activation of PCs had no effect on distance traveled, nor the time animals spent in the center of the arena, a measure of anxiety (*Figure 2e*). To assess the effect of stimulation on locomotion with greater temporal resolution, we averaged animal speed across all blocks of stimulation within the trial, centered around the onset of stimulation, and found that stimulation did not induce any transient change in locomotion (*Figure 2—figure supplement 1*). Similarly, optogenetic inhibition did not affect locomotion or the time animals spent in the center of the arena (*Figure 2f–g* and *Figure 2—figure supplement 1*). Together, these data suggest that manipulating vermal PC firing does not strongly affect coordination, locomotion or anxiety, although optogenetic stimulation of PCs at strong light intensities (greater than those used for behavioral assays) consistently drove profound motor impairment as previously described for vermal electrical stimulation (*Chambers, 1947*).

To assess the impact of cerebellar activity on aggressive and social behaviors we performed resident-intruder assays while optogenetically manipulating PC activity in the resident (aggressor) animal (*Figure 3a*). Although resident mice reliably display aggressive behaviors in resident-intruder assays, attacks occur at a relatively infrequent rate of <1 attack per minute (*Leypold et al., 2002*; *Yang et al., 2013*; *Lewis et al., 2015*). In order to increase baseline aggression, fiber-implanted mice were housed with females, providing the opportunity to mate, then subsequently singly-housed for at least 1 week prior to assays. Adult male BALB/c intruders were introduced into the resident's home cage for 10 min. Optogenetic stimulation was delivered to residents in alternating 1 min blocks. The onset and duration of multiple behaviors were recorded, including aggression (attacks, tail rattles, chasing and lateral threat), social encounters (face-to-face contact and ano-genital sniffing), as well as self-grooming by the resident (*Figure 3b*).

Optogenetic activation of vermal PCs significantly decreased the number of attacks (p=0.003, two-tailed paired Student's *t*-test) (*Figure 3c*, see *Figure 3—figure supplement 1* for detailed statistics). Stimulation did not affect the frequency of social interactions (p=0.7), or the rate of tail rattles, chasing, or lateral threat, though it did increase the rate of self-grooming by the resident (*Figure 3—figure supplement 2*). An advantage of the optogenetic approach we have used is that it allows us to precisely determine the time course of the effect of stimulation on aggressive behavior with greater temporal resolution. We binned attacks in 10 s increments and averaged across alternating blocks at the onset of stimulation. Even though attacks are infrequent and stochastic, this

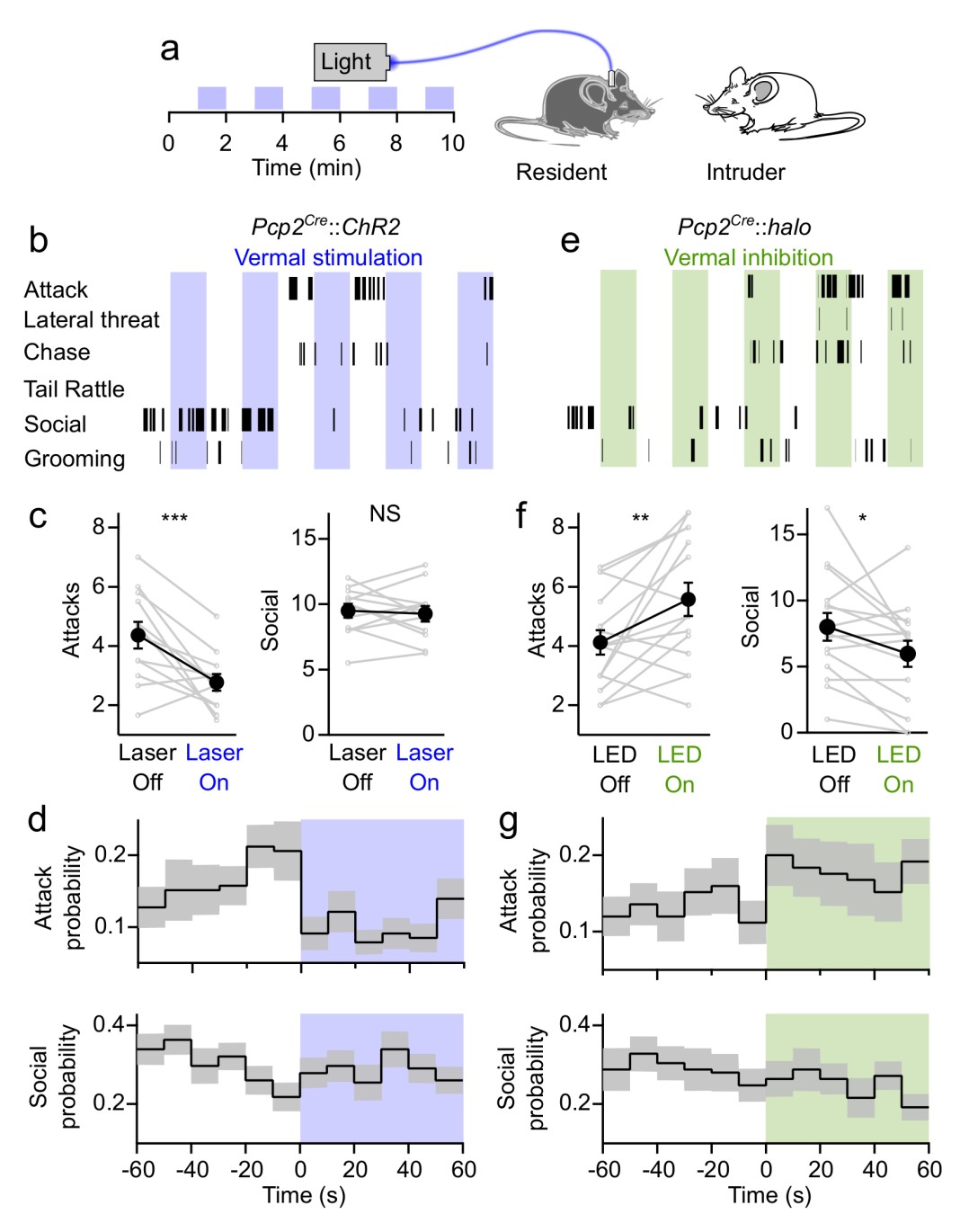

**Figure 3.** Bidirectional control of aggression by optogenetic modulation of vermal Purkinje cell activity. (**a**) Schematic for resident-intruder assays with optogenetic stimulation. (**b**) Representative scoring of social and aggressive behaviors. (**c**) Average number of attacks and social encounters during ChR2 assays (31 assays from 12 residents). (**d**) Peristimulus time histogram of the probability of attacks (top) and social investigations (bottom) within 10 s bins during epochs with and without ChR2-mediated excitation of vermal PCs. (**e–g**), Same as in (**b–d**), but during Halorhodopsin-mediated inhibition of vermal PCs (34 assays from 15 residents).

The online version of this article includes the following source data and figure supplement(s) for figure 3:

**Source data 1.** Resident Intruder Halo Vermis data.
**Source data 2.** Resident Intruder No Opsin Vermis data.
**Figure supplement 1.** Statistical significance of behavioral effects.
*Figure 3 continued on next page*

*Figure 3 continued*

**Figure supplement 2.** Effect of manipulating vermal Purkinje cell activity on grooming, tail-rattling, and aggressive lunging during resident-intruder assays.
**Figure supplement 3.** Aggression is not affected by light alone or by stimulating Crus II Purkinje cells.
**Figure supplement 3—source data 1.** Data for *Figure 3—figure supplement 3A*.
**Figure supplement 3—source data 2.** Data for *Figure 3—figure supplement 3B*.

analysis revealed that optical activation of PCs immediately reduced attack frequency, and when illumination was stopped the attack frequency gradually ramped up in the subsequent minute (*Figure 3d*). Stimulation reduced the frequency of attacks by 56% in the 10 s immediately following the onset of stimulation (*Figure 3d*). To put this into context, genetically ablating neurons in the ventromedial hypothalamus, a brain region colloquially referred to as the 'attack area' because of its importance in regulating aggression, decreases the attack frequency by a little more than 50% (*Yang et al., 2013*).

To test whether decreased attacks might result from a distracting influence of light escaping from the implanted optical fiber, we performed resident-intruder assays with a separate cohort of wild-type mice that did not express opsins, and found that optical stimulation had no effect on either attacks or social interactions (*Figure 3—figure supplement 3*). To test whether the effect on aggression was specific to stimulating activity in the posterior vermis, we repeated the experiments in ChR2-expressing animals but implanted the optical fiber over a region outside the vermis. Optogenetic manipulations of Purkinje cell activity in many regions have been shown to drive motor movements in mice (*Witter et al., 2013*; *Heiney et al., 2014*; *Proville et al., 2014*; *Lee et al., 2015*), which could indirectly affect the expression of aggression and other behaviors. To avoid the confound of motor effects, implants were positioned over Crus II, a region that has not been implicated in regulating aggression, but where manipulation of neuronal activity does not drive overt motor phenotypes in rodents (*Yamaguchi and Sakurai, 2016*). In these mice, stimulating PC firing had no effect on the frequency of attacks (*Figure 3—figure supplement 3*). Together, these results suggest that increased PC firing in the cerebellar vermis results in a rapid and significant decrease in aggression.

If elevating PC firing in the vermis decreases aggression, does suppressing PC firing increase aggression? It is not possible to address this question with electrical stimulation, but it is possible using optogenetics. Inhibiting PCs with halorhodopsin (*Figure 3e*) had opposing effects on aggression, significantly increasing the number of attacks (p=0.01), and decreasing social interactions (p=0.03) (*Figure 3f*). Averaging the attack frequency across multiple epochs of stimulation showed that attack frequency nearly doubled in the 10 s following the onset of halorhodopsin-driven inhibition of PCs (*Figure 3g*). These results indicate that the activity of PCs in the cerebellar vermis exerts a bidirectional influence over aggressive behavior.

## Discussion

Here we demonstrate that Purkinje cell activity in the posterior vermis drives rapid, bidirectional changes in aggressive behavior. Several aspects of our study provide important advances over previous studies that implicated the cerebellum in the regulation of aggression. First, we demonstrate that cerebellar activity regulates rodent aggression in an established assay that is amenable to quantification. This approach opens the door for quantitative studies in a genetically-manipulatable animal model, and promises to be beneficial for future studies of cerebellar control of aggression. Second, given the role of the cerebellum in motor control, it was important to determine whether the effects on aggression were a secondary consequence of impaired motor function. We evaluated this using open field and rotorod assays, and found that the same stimulation that altered aggression did not affect motor performance. Previous studies did not perform such a quantitative evaluation of motor performance. Third, the stimulation we used to suppress behavior was more selective than could be achieved with the electrical stimulation employed in previous studies, which in addition to stimulating PCs directly, can activate modulatory fibers, mossy fibers, climbing fibers, and inhibitory neurons in the cerebellar cortex. Consequently, in our ChR2 experiments, we can attribute decreased aggression to an increase in PC activity. Fourth, we showed that in our experiments

increases in PC firing decreased firing in the DCN and decreasing PC firing did the opposite. Fifth, we find that suppressing PC activity increased aggression. This is consistent with the observation that stimulating the DCN promotes aggression (*Zanchetti and Zoccolini, 1954*; *Reis et al., 1973*), but differs from the observation that cerebellar lesions suppress aggression. Studies in other brain regions have described similar behavioral differences between the effects of acutely manipulating activity and lesions (*Hong et al., 2018*). It is likely that lesions within the vermis are accompanied by compensatory mechanisms within the DCN (*Bäurle et al., 1997*). Finally, by optogenetically regulating PC firing, we provide evidence that the cerebellum can rapidly and bidirectionally regulate aggression.

The present study raises a number of important questions regarding the manner in which the cerebellum controls behavior. What specific region of the cerebellar cortex is involved? We find that manipulating the activity of Purkinje cells in the posterior vermis is sufficient to significantly modulate aggression. This is consistent with clinical studies implicating lobule VII of the vermis in affective processing (*Stoodley and Schmahmann, 2009*). However, it is difficult to determine the precise region of the cerebellum that was influenced by our optogenetic stimulation. Light emanating from the face of implanted fibers scatters and disperses over hundreds of microns (*Aravanis et al., 2007*; *Li et al., 2019*). Consequently, it is likely that in addition to lobule VII, lobules VIb and VIII received significant illumination, and that PC activity in these regions was also likely modulated to some extent. More detailed studies that manipulate activity in other areas of the midline vermis could add clarity to the specific regions of the cerebellar cortex that regulate aggression. It is also possible that more specific regulation of PC activity in the region controlling aggression (for example, without affecting PC firing in neighboring regions that alter other behaviors) will lead to larger effects on aggression. This could be accomplished by performing similar experiments to those described here, but using viral expression of ChR2 or halorhodopsin to restrict optogenetic manipulation to specific regions of the cerebellar cortex. Furthermore, what is the nature of inputs that control this region? Different regions of the cerebellar cortex typically combine mossy fiber inputs from diverse sources, and it will be interesting to determine how these inputs are combined within the cerebellum to control aggression. Finally, what is the output pathway and the downstream targets that are ultimately regulated by activity in this region of the cerebellar cortex? Anatomical studies have described connections between the cerebellum and regions implicated in aggression, including hypothalamus (*Haines et al., 1997*) and prefrontal cortex (*Kelly and Strick, 2003*; *Suzuki et al., 2012*). Electrophysiological recordings have found that cerebellar stimulation evokes responses in those regions, along with limbic structures such as the hypothalamus, amygdala, and hippocampus (*Anand et al., 1959*; *Snider and Maiti, 1976*). Yet, while the somatotopic organization of the cerebellum is well characterized in regions that influence motor function, the output pathways of areas like the posterior vermis have yet to be clearly defined.

It is interesting to speculate on the nature of the role of the cerebellum in controlling aggressive behavior. The cerebellum has expanded in size relative to the cerebral cortex over the course of human evolution (*Weaver, 2005*), it contains more than half the neurons in brain and it possesses myriad connections to other brain regions. It is unsurprising that its influence should extend beyond the motor realm. Experiments on motor control suggest that the cerebellum combines inputs to generate predictions. It is natural to think that this computational strategy might be used by the posterior vermis of the cerebellum to learn how to respond to cues, and to ultimately decide when aggression is the correct response. Perhaps even subtle dysfunctions or misdirected plasticity within this region can lead to inappropriate aggressive behavior. For example, cerebellar damage often occurs in patients with PTSD (*Rabellino et al., 2018*). As non-invasive stimulation techniques like transcranial magnetic stimulation of the cerebellum emerge as a clinical treatment options (*Demirtas-Tatlidede et al., 2010*), it is increasingly important to understand the which areas of the cerebellum control non-motor behaviors (*Kelly and Strick, 2003*). Future work could shed light on the anatomical projections and physiological impact of non-motor regions of the cerebellum.

## Materials and methods

### Animals

All experiments were conducted in accordance with federal guidelines and protocols approved by the Harvard Medical Area Standing Committee on Animals. Male mice of the following strains were used: Resident mice were either wild-type (WT) C57BL/6N (Charles River Laboratories), or Pcp2-cre mice (Jackson Laboratory, stock number 010536) crossed to either ChR2-EYFP (Ai32, Jackson Laboratory, 024109) or eNpHR3.0-EYFP (Halo) mice (Ai39 Jackson Laboratory, 014539). Intruder mice were BALB/c (Charles River Laboratories).

### In vitro physiology

Sagittal cerebellar slices were prepared from adult mice (P30-P100) and recordings were performed as previously described (*Jackman et al., 2014*). Briefly, animals were anesthetized with isoflurane and euthanized by decapitation. Brains were removed into oxygenated ice-cold cutting solution containing (in mm): 82.7 NaCl, 65 sucrose, 23.8 NaHCO$_3$, 23.7 glucose, 6.8 MgCl$_2$, 2.4 KCl, 1.4 NaH$_2$PO$_4$, and 0.5 CaCl$_2$. Sagittal slices from the cerebellar vermis (250 µm thick) were prepared in ice-cold cutting solution using a Leica VT1200s vibrotome. Slices were transferred for 30 min into oxygenated artificial CSF (ACSF) at 32°C containing the following (in mm): 125 NaCl, 26 NaHCO$_3$, 25 glucose, 2.5 KCl, 2 CaCl2, 1.25 NaH$_2$PO$_4$, and 1 MgCl$_2$, adjusted to 315 mOsm, and allowed to equilibrate to room temperature for >30 min prior to recording. PCP2-Cre::ChR2-EYFP were used for all ChR2 recordings. Halorhodopsin recordings were performed in PCP2-Cre mice where opsin expression was driven by stereotaxic cerebellar injections (as previously described [*Jackman et al., 2014*]) of AAV9.EF1a.DIO.eNpHR3.0-eYFP.WPRE.hGH (Addgene26966). Although these mice were not used for behavioral experiments, similar optical sensitivity was observed in recordings performed for a separate study using PCP2-Cre::Ai39 mice (*Guo et al., 2016*).

Data were acquired using a Multiclamp 700B amplifier (Molecular Devices) digitized at 10 kHz with an ITC-18 (Instrutech), and low-pass filtered at 4 kHz. Acquisition and analysis were performed with custom software written in IgorPro (generously provided by Matthew Xu-Friedman, SUNY Buffalo). Whole-cell current clamp or on-cell recordings were obtained using borosilicate patch pipettes (2–4 MΩ), the internal solution contained the following (in mm): 150 K-gluconate, 3 KCl, 10 HEPES, 0.5 EGTA, 3 MgATP, 0.5 GTP, five phosphocreatine-tris2, and five phosphocreatine-Na2, with the pH adjusted to 7.2 with NaOH. Optical stimulation was delivered through the excitation pathway of a BX51WI upright microscope (Olympus) by either a 50 mW DPSS analog-controllable 473 nm blue laser (MBL-III-473–50 mW, Optoengine), or a 590 nm Amber LED (160 mW, ThorLabs).

### Chronic fiber implantation and in vivo stimulation

Optical fiber implants were assembled as previously described (*Sparta et al., 2012*). Briefly, a multi-mode optical fiber (Thorlabs, NA 0.39, 200 µm core) was secured into ceramic ferrules (Thorlabs, 1.25 mm O.D.) with epoxy. Fibers were cleaved to protrude 0.2 mm below the ferrule, and the connector end was polished. Only fibers with >70% transmissivity were used. To determine the intensity of light exiting fibers, the output of fibers was measured with a power meter (Ophir; Vega). A photodiode was used to measure the relative intensity during short flashes controlled by the analog trigger of the laser, and this value was used to compute the power output during short flashes. The intensity of light delivered in vivo was computed by dividing the total light output (4.1 mW for the 473 nm laser, 2.3 mW for the 590 nm LED) by the surface area of the optical fiber. To avoid desensitizing ChR2 and driving PCs into depolarization-induced block, we stimulated ChR2-expressing animals using flash trains of 1–2 ms pulses. Optical fibers were implanted as described previously (*Sparta et al., 2012*). Briefly, adult mice (P40–P80) were anaesthetized with ketamine/xylazine (100/ 10 mg/kg) supplemented with isoflurane (1%–4%). An incision was performed to expose the skull, and the connective tissue and musculature above the cerebellum was gently peeled back. The stereotaxic coordinates used to target implants to the posterior vermis and Crus II were initially determined by performing test craniotomies, and injecting small volumes of fluorescent dye or viral expression vectors. Animals were sacrificed, and posthoc fluorescence microcopy was used to determine the location of craniotomies relative to the cerebellar surface. For vermal implants, the site for the craniotomy was determined using a fine pipette attached to a stereotaxic device (Kopf). After

locating bregma, the pipette was moved caudal to the cerebellum, lowered 2.0 mm relative to bregma, then advanced rostrally until it touched the surface of the exposed skull. The site of Crus II craniotomies were determined similarly, but 1.5 mm ventral and 2.5 mm lateral of bregma. A craniotomy was performed at this site, and implants were lowered into place. Implants were secured to the skull using Metabond (Parkell), and the wound was sutured. Buprenorphine (0.05 mg/kg) was post-operatively administered subcutaneously every 12 hr for 48 hr. At the conclusion of behavioral assays, some resident animals were anaesthetized with ketamine and transcardially perfused with 4% paraformaldehyde (PFA) in PBS. The brain was removed, post-fixed for 24 hr, and the sagittal cerebellar slices were prepared using a vibratome. YFP fluorescence in Ai32 and Ai39 mice was imaged using an Olympus MVX10 Macro dissecting microscope (for intact fixed tissue) or Zeiss Axio Imager (for brain slices). Images were contrast enhanced in Fiji for visualization. In some cases, histology could be used to determine the location of implant over the cerebellar cortex.

## In vivo physiology

Mice from behavioral experiments were heavily anesthetized with isoflurane (2%). Anesthesia was maintained for all following procedures. A craniotomy immediately lateral to the implanted optical fiber was made to insert an electrode for extracellular recordings. A headplate was cemented (Metabond) anterior to the optical fiber, and the mouse was head-fixed for recordings. Electrodes were pulled from borosilicate glass (Sutter), filled with ACSF, and were inserted at an angle between 20 and 45 degrees to record single unit activity below the optical fiber. Most neurons were recorded between 1 and 2 mm from this entry point. Signals were acquired at 20 kHz between 0.2 and 7.5 kHz (Intan Technologies). Purkinje cells were identified by the presence of complex spikes, characteristic increase in noise as the electrode entered the Purkinje cell layer, and/or responsiveness to light. Single units were the sorted offline in Offline Sorter (Plexon) and analyzed in Matlab (Mathworks).

Recordings from the DCN were made in awake, head-restrained PCP2-Cre::ChR2-EYFP and PCP2-Cre::Halo-EYFP mice. Briefly, mice were anesthetized with isoflurane (2%) and a headplate was cemented (Metabond) onto the anterior aspect of the cranium. A craniotomy was made (−6 mm posterior to bregma, 0.75–1.2 mm lateral from the midline) for placement of the recording electrode into the medial or interposed DCN. For PCP2-Cre::ChR2-EYFP mice, a large craniotomy was made over the vermis to facilitate activation of Purkinje cells. For PCP2-Cre::Halo-EYFP mice, the region over the vermis was manually thinned till visibly transparent under a dissecting scope. Optical stimulation was delivered via an optical fiber positioned above the vermis. Light was focused to a spot size of approximately 2 mm diameter. Stimulation parameters were identical to those used in anesthetized experiments above, though in lieu of LEDs, these experiments used either a 473 nm blue laser (MBL-III-473–50 mW, Optoengine) for ChR2 activation, or a 647 nm red laser (MRL-III-635–500 mW, Optoengine) for Halo activation. Recordings were made using a silicon probe (E-series, Cambridge Neurotech), from awake mice head-restrained over a cylindrical treadmill. DCN were identified by the recording depth, absence of Purkinje cell activity, and in a subset of experiments, the silicon probe was coated with DiI (Thermo Fisher Scientific) to confirm that the recording sites were within the DCN. Data were acquired and processed identically to anesthetized experiments.

## Behavior

Mice used in behavioral experiments were housed in a 12 hr reverse light-dark cycle (lit 7PM-7AM). The timeline for experiments were as follows: Resident (aggressor) mice were allowed to recover from implants surgeries for at least 7 days. They were then paired with an adult C57BL/6N female for 7–12 days. The female was removed and the resident mouse remained in social isolation for at least 7 days. No cage changes were performed during social isolation to enhance subsequent territorial dominance aggression. Residents were first tested for signs of stimulation-induced motor dysfunction, then assayed in the open field, rotarod, and finally aggression (resident-intruder) over the course of several weeks. Prior to behavioral experiments, animals were placed in a darkened room and allowed to habituate for at least 1 hr.

For experiments involving optogenetic stimulation the light source was connected to a fiberoptic cable via a rotating commutator (FRJ_1 × 1_FC-FC, Doric Lenses) to allow freedom of motion. The fiberoptic cable was attached to the implant with a ferrule sleeve (Thorlabs) and mice were allowed to acclimate to the attached cable for 30 min. All assays were conducted under dim red illumination.

Sensorimotor coordination was assessed with an automated rotarod apparatus (UgoBasile). Mice were placed on a rotarod with a constant rotation of 4 RPM, and allowed to acclimate for 1 min, after which the rod accelerated to 60 RPM at a rate of 20 RPM/min. Time to fall was calculated from the beginning of acceleration. All mice were run on two consecutive trials with 4 min rest between trials, and animals were randomly assigned to receive optical stimulation during either the first or second trial. Optical stimulation began 10 s before the onset of acceleration and continued until the animal fell. Open field assays were conducted in a square opaque white plastic container (46 × 46 cm), and the central regions was defined as a square one third the dimension of the area. Automated tracking was performed in Matlab using idTracker2.1.

For resident-intruder assays, residents were attached to the optical fiber and allowed to acclimate in their home change for 30 min. A BALB/c intruder (roughly age matched) was introduced into the home cage, and interactions were filmed and manually scored. Trials were stopped in the event that either animal was injured by an attack, or if the resident attacked continuously for more than 60 s. Residents were run on up to five resident-intruder assays with at least 2 days between assays, with a novel intruder used for each assay. Residents were removed from the study if they failed to attack, or if the intruder attacked (*Leshner and Nock, 1976*). To establish a baseline level of aggression, assays with less than three attacks or more than 20 attacks were omitted from analysis. A subset of halorhodopsin-expressing animals (4/15) were stimulated in 5 min intervals rather than the standard 1 min intervals. Resident intruder assays were scored manually by an experimenter blinded to mouse genotype and stimulation wavelengths, and annotated using the open-source software BORIS (*Friard and Gamba, 2016*). The following behaviors were scored; self-grooming by the resident, social interactions (including face-to-face contact, mutual grooming and ano-genital sniffing), tail-rattles, lateral threat, chasing of the intruder by the resident, and biting attacks (*Koolhaas et al., 2013*).

Data analysis was performed using Igor Pro (Wavemetrics). All results are expressed as mean ± standard error of mean. Data for each behavior were tested for normality for each experimental condition (Shapiro-Wilk test). To compare light on/light off differences within groups, data that did not meet the criterion for normality were analyzed by non-parametric Wilcoxon signed-rank, while normally distributed data were analyzed by two-tailed paired Student's *t*-test. Similarly, comparisons between groups were tested by two-tailed unpaired Student's *t*-test, or the non-parametric Wilcoxon-Mann-Whitney test. The criterion for statistical significance was set at $p < 0.05$.

## Acknowledgements

This work was supported by the GVR Khodadad Research Fund for the Study of Excessive (Pathological) Selfishness and Aggressive Behavior. We especially wish to thank Ghahreman Khodadad for both his financial support and stimulating discussions. We also thank Jasmine Vazquez for illustrations, and Michelle Ocana and the Neurobiology Imaging Center for help with microscopy. This facility is supported in part by the Neural Imaging Center as part of a NINDS P30 Core Center grant (NS072030). This work was supported by a Nancy Lurie Marks postdoctoral fellowship to SLJ, a NIH postdoctoral fellowship F32NS101889 to CHC, and NIH grant NINDS R35NS097284 to WGR.

## Additional information

### Funding

| Funder | Grant reference number | Author |
| --- | --- | --- |
| NIH Office of the Director | R35NS097284 | Wade G Regehr |
| The Khodadah Research Fund | | Wade G Regehr |
| NIH Office of the Director | F32NS101889 | Christopher H Chen |
| Nancy Lurie Marks Family Foundation | Postdoctoral fellowship | Skyler L Jackman |

The funders had no role in study design, data collection and interpretation, or the decision to submit the work for publication.

## Author contributions
Skyler L Jackman, Conceptualization, Supervision, Investigation, Methodology, Writing - original draft; Christopher H Chen, Investigation, Writing - original draft; Heather L Offermann, Iain R Drew, Bailey M Harrison, Anna M Bowman, Katelyn M Flick, Isabella Flaquer, Investigation; Wade G Regehr, Conceptualization, Resources, Supervision, Funding acquisition, Writing - original draft

## Author ORCIDs
Skyler L Jackman ⓘD https://orcid.org/0000-0002-6500-3937
Wade G Regehr ⓘD https://orcid.org/0000-0002-3485-8094

## Ethics
Animal experimentation: All experiments were conducted in accordance with federal guidelines and protocols (#1493) approved by the Harvard Medical Area Standing Committee on Animals.

## Decision letter and Author response
Decision letter https://doi.org/10.7554/eLife.53229.sa1
Author response https://doi.org/10.7554/eLife.53229.sa2

---

# Additional files

## Supplementary files
• Transparent reporting form

## Data availability
Source data files have been provided for Figures 1, 2 and 3.

---

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
