## [Decision Letter]

**Acceptance summary:**

Though the cerebellum is known to mediate motor co-ordination and balance, it is also a centre for non-motor functions. Using the power of optogenetics to specifically manipulate activity bidirectionally in cerebellar Purkinje neurons, this study draws a connection between the cerebellum and aggression. These highly interesting results now pave the way for future studies on how Purkinje neurons regulate aggression.

**Decision letter after peer review:**

Thank you for submitting your article "Cerebellar Purkinje cell activity modulates aggressive behavior" for consideration by *eLife*. Your article has been reviewed by three peer reviewers, one of whom is a member of our Board of Reviewing Editors, and the evaluation has been overseen by Richard Ivry as the Senior Editor. The reviewers have opted to remain anonymous.

The reviewers have discussed the reviews with one another and the Reviewing Editor has drafted this decision to help you prepare a revised submission.

Jackman et al. show that optogenetic stimulation of Purkinje neurons in the posterior vermis reduces aggressive behavior while optogenetic suppression increases it. While it has been known that the cerebellum is important for non-motor functions including aggression, the authors' use of reversible, optogenetic manipulations make their results compelling.

However, there are several concerns that were raised by the reviewers and these are listed below.

Essential revisions:

1) A central concern is that the authors' results run contrary to decades of studies showing that cerebellar lesions "tame" animals. Some of these studies are cited by the authors. Thus, it is quite surprising that the authors don't comment on the disparity between their results and those of many prior investigators.

2) Optogenetic effects on Purkinje cell firing may not be as straightforward as the authors suggest, as is outlined in the Introduction of Streng and Krook-Magnuson (2020, J. Physiol.). Perhaps this disparity can be explained by differences in the application of the optogenetic technique between studies. However, the disparity between the present results and those of lesions studies makes one wonder.

3) Results section:

"Fibers were positioned at the midline over lobule VII (Figure 1).… "

The authors indicate that they are illuminating vermal lobule VII, however they don't provide any concrete evidence that this is the case. The authors should show specific histology that indicates the location of the optical fiber. They also don't indicate whether their illumination (and its effects) were confined to lobule VII or involved other nearby cerebellar lobules as well. The authors need to analyze and describe the areal extent of P-cell activation and inactivation they caused. The authors selected vermal lobule VII based on reviews of the human neuroimaging literature. Others have found cerebellar effects on the regulation of emotion to be located in slightly more anterior vermal lobules. The authors achieved effects, but it isn't clear that the authors were illuminating the right or optimal vermal lobule. Related to this, the authors should at least discuss the modeled/projected scope of activation/inhibition predicted by the light power used in these studies.

4) It would have been comforting for the authors to demonstrate that optogenetic manipulation could affect motor behavior at other sites in the cerebellum. Then, the authors could have used the same parameters of optogenetic stimulation that were effective for motor behaviors to test the effects on non-motor behavior in vermal lobule VII.

5) The absence of a change in "anxiety" is somewhat troubling given the extensive literature on the role of the cerebellar vermis in fear conditioning (e.g., Sacchetti et al.).

[Editors' note: further revisions were suggested prior to acceptance, as described below.]

Thank you for submitting your article "Cerebellar Purkinje cell activity modulates aggressive behavior" for consideration by *eLife*. Your article has been reviewed by two reviewers, one of whom is a member of our Board of Reviewing Editors, and the evaluation has been overseen by Richard Ivry as the Senior Editor. The reviewers have opted to remain anonymous.

Though we find the work potentially exciting, and agree that the revision is a significant improvement over the original submission, we are disappointed that the revised version has not satisfactorily answered all of the concerns raised by the reviewers. These are significant concerns regarding the specificity of the manipulations and the interpretation of the results. They are listed again below. We hope that you can resolve these issues in the next version of the manuscript.

1) From the previous decision letter:

"The authors indicate that they are illuminating vermal lobule VII, however they don't provide any concrete evidence that this is the case. The authors should show specific histology that indicates the location of the optical fiber."

This point was not addressed in the revision nor in the response. The absence of this data leads to considerable confusion. For example, Figure 2B shows the authors' designation of vermal lobules. Figure 1A illustrates the location of an optogenetic probe and a recording electrode. A comparison of Figure 2B with Figure 1A suggests that the probe and recordings were made in lobule VIa, not VII. Thus, the authors leave the reader with considerable uncertainty and confusion regarding the site of their effects.

2) The authors could have performed the control experiment of using optogenetic activation/inactivation of the classical motor territory of the vermis. This would have allowed them to determine the appropriate activation and inactivation thresholds and parameters to produce motor effects. This approach would have allowed them to make clean statements about optogenetic activation/inactivation relative to the classical motor symptoms of cerebellar cortical lesions/stimulation. Then, the authors could have used the same parameters to stimulate the posterior vermis. The absence of this "motor control" remains a significant shortcoming of the present study.

3) Classical studies vs. optogenetics: The authors have dug through the classical literature in an impressive fashion. However, the effects of optogenetic stimulation may be no less complex to interpret than lesions of the cerebellar cortex. For example, though the authors show a reduction in DCN firing rates after PC activation, and claim that PC's are not synchronized, no data is shown to support this claim and a role for synchrony is not ruled out.

In light of 2 and 3 above, authors should tone down their claims such as the Abstract that "These results *establish* Purkinje cell activity in the cerebellar vermis regulates aggression…" (emphasis added). It would be most appropriate to replace "establishes," wherever it appears, with "supports."

---

## [Author Response]

Essential revisions:1) A central concern is that the authors' results run contrary to decades of studies showing that cerebellar lesions "tame" animals. Some of these studies are cited by the authors. Thus, it is quite surprising that the authors don't comment on the disparity between their results and those of many prior investigators.

The reviewer raises an interesting point about discrepancies in cerebellar studies of aggression. These discrepancies predate our own studies. In experiments performed on humans, monkeys and cats, stimulation of the vermis and lesions of the vermis both reduce aggression. As Purkinje cells of the vermis inhibit cells in the deep cerebellar nuclei, stimulation and ablation of Purkinje cells would be expected to have opposite effects. But as we point out, there are limitations to both approaches. Notably, the hypothesis that lesions of the cortex would increase activity in the DCN is not supported by experimental evidence. Past studies have shown that mouse models with near total degeneration of PCs produce compensatory changes in DCN neurons that paradoxically lead to *lowered* DCN firing rates. We now reference one such study (Baurle et al., 1997). We have now expanded the Introduction to more clearly point out the discrepancy in the sign of the effect of the cerebellar vermis on aggression in the previous literature. We have also included a discussion of the approach we have taken and how the bidirectional specific manipulation of PC activity allows us to be confident about the effects of PC activity on aggression.

2) Optogenetic effects on Purkinje cell firing may not be as straightforward as the authors suggest, as is outlined in the Introduction of Streng and Krook-Magnuson (2020, J. Physiol.). Perhaps this disparity can be explained by differences in the application of the optogenetic technique between studies. However, the disparity between the present results and those of lesions studies makes one wonder.

As stated in response to point 1, our results agree with previous stimulation experiments and differ from lesion studies, and we have discussed this in the text. The reviewer is concerned about the use of optogenetics and suggests that perhaps we could be getting an artifact because of complications in the use of optogenetics. The paper cited by Streng and Krook-Magnuson (2020) does not assess the optogenetic activation of Purkinje cells, but instead summarizes the literature to motivate their approach of influencing the deep nuclei directly. Based on our reading of the literature in which PCs are manipulated with optogenetics, and our own measurements of PC responses to optogenetic manipulations, we do not share the concerns of the reviewer. Another important issue is the issue of pauses promoting PC firing. We are very familiar with this issue and have a paper under review dealing with pauses in PC firing and DCN excitability. Based on Person and Raman, (2011), synchronous activation of PCs could promote firing in the DCN. However, based on our recordings of light-evoked PC firing, and the synchrony required to promote firing, we conclude that the optogenetic approach we have used does not lead to synchronous PC firing that is sufficiently precise to promote pauses and increases in DCN firing.

Most importantly, we also assess the effect of optogenetic manipulations on the firing of DCN neurons. These new experiments were performed in awake animals. We found that increasing PC firing with ChR2 decreased the firing of DCN neurons, and decreasing PC firing with halorhodopsin increased the firing of neurons in the DCN. We have added 3 panels to Figure 1 describing these new experiments.

3) Results section:"Fibers were positioned at the midline over lobule VII (Figure 1).… "The authors indicate that they are illuminating vermal lobule VII, however they don't provide any concrete evidence that this is the case. The authors should show specific histology that indicates the location of the optical fiber. They also don't indicate whether their illumination (and its effects) were confined to lobule VII or involved other nearby cerebellar lobules as well. The authors need to analyze and describe the areal extent of P-cell activation and inactivation they caused. The authors selected vermal lobule VII based on reviews of the human neuroimaging literature. Others have found cerebellar effects on the regulation of emotion to be located in slightly more anterior vermal lobules. The authors achieved effects, but it isn't clear that the authors were illuminating the right or optimal vermal lobule. Related to this, the authors should at least discuss the modeled/projected scope of activation/inhibition predicted by the light power used in these studies.

The reviewer correctly points out that the approach we have taken has limitations with regard to precise localization of the region of the posterior vermis that controls aggression. In the original manuscript we avoided making claims about the precise region involved in regulating aggression. We are now more explicit and point out other approaches that would be better suited to the question of localization.

4) It would have been comforting for the authors to demonstrate that optogenetic manipulation could affect motor behavior at other sites in the cerebellum. Then, the authors could have used the same parameters of optogenetic stimulation that were effective for motor behaviors to test the effects on non-motor behavior in vermal lobule VII.

Due to the extensive literature describing the motor functions of the cerebellum and optogenetic perturbation of movements (Witter et al., 2013; Lee et al., 2015), we did not attempt to replicate previous experiments. We agree that in the interest of thoroughness the experiments the reviewer proposes would have provided additional context to our study. However, we observed motor effects during strong stimulation of the vermis (similar to those described by several older studies). We have added additional descriptions of these motor effects and our efforts to avoid them during behavioral assays. For our control experiments, we chose not to manipulate a region with known influence over rodent motor coordination, because it would have presented a confound to our aggression assays. For this reason we chose to stimulate Crus II, a region that does not appear to affect coordination in rodents (Yamaguchi and Sakurai, 2016) but which has not been implicated in regulating aggression.

5) The absence of a change in "anxiety" is somewhat troubling given the extensive literature on the role of the cerebellar vermis in fear conditioning (e.g., Sacchetti et al.).

We suspected that stimulating the cerebellar vermis might lead to decreased measures of anxiety in the open field, because human patients receiving chronic vermal stimulation reported decreased levels of anxiety. However, we saw no effect on anxiety as measured by open field. We do not view this as troubling: several past studies that reported a role for the vermis in fear conditioning and the consolidation of emotional memories also performed open field assays, and reported no effect on either locomotion or anxiety in the open field (Sacchetti et al., 2002; Sacchetti et al., 2004; Koutsikou et al., 2015)

[Editors' note: further revisions were suggested prior to acceptance, as described below.]

Though we find the work potentially exciting, and agree that the revision is a significant improvement over the original submission, we are disappointed that the revised version has not satisfactorily answered all of the concerns raised by the reviewers. These are significant concerns regarding the specificity of the manipulations and the interpretation of the results. They are listed again below. We hope that you can resolve these issues in the next version of the manuscript.

We did our best to respond to the initial review, and we are pleased that we were able to successfully address most of the initial concerns. We were disappointed that we were unable to address three issues, but we hope we have successfully addressed them now.

1) From the previous decision letter:"The authors indicate that they are illuminating vermal lobule VII, however they don't provide any concrete evidence that this is the case. The authors should show specific histology that indicates the location of the optical fiber."This point was not addressed in the revision nor in the response. The absence of this data leads to considerable confusion. For example, Figure 2B shows the authors' designation of vermal lobules. Figure 1A illustrates the location of an optogenetic probe and a recording electrode. A comparison of Figure 2B with Figure 1A suggests that the probe and recordings were made in lobule VIa, not VII. Thus, the authors leave the reader with considerable uncertainty and confusion regarding the site of their effects.

We are sorry that we failed to address this issue satisfactorily in our initial response, and we now cover this topic extensively. First, we changed the schematic in Figure 1A, which incorrectly implied that our in vitro recordings targeted lobule VIb. The schematic now clearly depicts recordings from lobule VII in a brain slice that is viewed from above. Second, in the Materials and methods section we now describe the initial targeting strategy we used to determine the coordinates of our implants. In brief, we performed a series of preliminary dye/viral injections to determine the proper location of craniotomies over lobule VII. In the image presented here we provide the reviewers with an example of halorhodopsin-YFP expression in lobule VII (with some expression in lobule VIb) following injection of a cre-dependent AAV into PCP2-cre mice (Author response image 1). Moreover, we frequently sacrificed animals after behavioral testing and obtained fluorescence images of whole mouse brains. In some cases, it was possible to infer the location of the implant due to subtle blemishes on the cortical surface. An example is provided in Author response image 1.

We were previously careful to point out that our approach could not be used to determine the precise location of the cerebellar vermis that regulated aggression. We have extended this discussion of the caveats that do not allow us to definitively ascribe our behavioral effect to lobule VII. In our animals, ChR2/halorhodopsin are expressed in all areas of the cerebellar cortex, and lobule VII is only ~500 µm across. Thus, it is possible that implants positioned over lobule VII also stimulate lobule VIb and lobule VIII.

2) The authors could have performed the control experiment of using optogenetic activation/inactivation of the classical motor territory of the vermis. This would have allowed them to determine the appropriate activation and inactivation thresholds and parameters to produce motor effects. This approach would have allowed them to make clean statements about optogenetic activation/inactivation relative to the classical motor symptoms of cerebellar cortical lesions/stimulation. Then, the authors could have used the same parameters to stimulate the posterior vermis. The absence of this "motor control" remains a significant shortcoming of the present study.

As stated in the initial manuscript and in our initial response, we found that comparable stimulation of another non-motor cerebellar region did not alter aggression. We think this is an appropriate control experiments that is not confounded by motor effects. The reviewer suggested the alternative strategy, which was to stimulate a motor region. We feel that such a strategy is similar to the one we have provided, with the advantage that a motor effect establishes that the region has been stimulated, but with the disadvantage that the motor effect is very likely to confound the results of our behavioral assays. Our in vivo recordings provide evidence that our optogenetic stimulation modulates the activity of PCs below our implanted fibers. Moreover, optogenetic stimulation of Purkinje cells in other cerebellar regions has been shown to drive motor effects repeatedly by other groups (Heiney et al., 2014; Lee et al., 2015; Proville et al., 2014; Witter et al., 2013; Ten Brinke et al., 2017). Thus, for the reasons set forth in our previous response, we do not consider this issue a significant shortcoming of the paper. We have provided additional clarification in the Results section.

It should also be pointed out that it would take months to perform these additional control experiments. There is the added complication that our lab is shut down for at least the next 6-8 weeks, and in all likelihood much longer. Our mouse colonies have been reduced to very low levels and it would take considerable time after the lab reopens to have the mice available for additional experiments. In the best case, it would take us more than 6 months to do the requested experiments, and in all likelihood much longer.

We therefore hope the reviewers and editors will reconsider the requirement of stimulation of a motor region of the cerebellar cortex as a control experiment.

3) Classical studies vs. optogenetics: The authors have dug through the classical literature in an impressive fashion. However, the effects of optogenetic stimulation may be no less complex to interpret than lesions of the cerebellar cortex. For example, though the authors show a reduction in DCN firing rates after PC activation, and claim that PC's are not synchronized, no data is shown to support this claim and a role for synchrony is not ruled out.

In the initial review there was concern about potential complications of optogenetic stimulation. We were under the impression that the reviewer was concerned about PC synchronization based on the studies and Person and Raman, who found that synchronous PC firing can paradoxically elevate the firing frequency of DCN neurons. Consequently, it was unclear if our optogenetic PC activation would lead to sufficiently synchronous firing that DCN firing would be increased, or whether the overall increase of inhibition would suppress DCN firing.

Our in vivo recordings from the DCN show that optogenetic stimulation of PCs produced a net decrease in DCN activity (Figure 1H). Nevertheless, we now analyze our recordings on millisecond time scales to determine (1) whether brief light flashes presented at 20 Hz synchronize PC activity, (2) if there is a pause in PC firing following each light flash, and (3) if PC synchrony drives synchronous rebound firing in DCN cells. Our new analysis, presented in Figure 1—figure supplement 2, shows that although brief light flashes can evoke complex temporal responses in PCs, there is no discernible pause in average PC firing, and the net effect is suppression of DCN activity. This indicates to us that optogenetic stimulation of PCs did not elevate the firing of DCN neurons by promoting synchronous activity. In addition, we found that halorhodopsin can be used to reliably suppress PC firing which in turn increases firing in the DCN (Figure 1F,I). For these reasons we feel that we could use ChR2 to effectively stimulate PCs and suppress DCN firing, and use halorhodopsin to suppress PC firing and elevate DCN firing. We believe our recordings justify these conclusions.

In light of 2 and 3 above, authors should tone down their claims such as the Abstract that "These results establish Purkinje cell activity in the cerebellar vermis regulates aggression…" (emphasis added). It would be most appropriate to replace "establishes," wherever it appears, with "supports."

We have made this change.